# Hydrogel-Based Therapeutic Strategies for Periodontal Tissue Regeneration: Advances, Challenges, and Future Perspectives

**DOI:** 10.3390/pharmaceutics17111382

**Published:** 2025-10-24

**Authors:** Bowen Wang, Fengxin Ge, Wenqing Wang, Bo Wang, Cory J. Xian, Yuankun Zhai

**Affiliations:** 1School of Stomatology, Henan University, Kaifeng 475000, China; wangbw512@163.com (B.W.); 15893142513@163.com (F.G.); 15937820049@163.com (W.W.); 15037374833@163.com (B.W.); 2Kaifeng Key Laboratory of Periodontal Tissue Engineering, Kaifeng 475000, China; 3UniSA Clinical and Health Sciences, University of South Australia, Adelaide, SA 5001, Australia; cory.xian@unisa.edu.au

**Keywords:** periodontitis, hydrogel, periodontal tissue regeneration, scaffold, seed cells

## Abstract

Periodontitis, a prevalent chronic infectious disease triggered by oral biofilm microbiota, results in progressive destruction of periodontal supporting tissues, and conventional treatments have limited therapeutic effects on it. Hydrogels, due to their excellent biocompatibility, three-dimensional extracellular matrix-like structure, and localized sustained-release properties, can provide support for cell attachment, promote cell proliferation and differentiation, and improve drug utilization efficiency, showing great promise for applications in treating periodontitis as well as promoting periodontal tissue regeneration. This article first introduces the limitations of current periodontitis treatments and the unique advantages of hydrogels in periodontitis treatment and periodontal tissue regeneration, and then provides an overview of the classifications of hydrogels, the active substances they can load, and the characteristics and functions of these active substances. Subsequently, the article introduces the latest advances in the application of several common natural polymer hydrogels in periodontal tissue regeneration. Finally, the article discusses the current limitations of hydrogels in terms of structure and properties, and proposes potential solutions and future development directions in periodontal tissue regeneration.

## 1. Introduction

Periodontitis (PD), a prevalent chronic infectious disease, has emerged as a global public health concern, affecting approximately 20% to 50% of the population worldwide [1]. PD arises from the interplay between dental plaque biofilm and host immune responses, which leads to progressive destruction of the periodontal ligament (PDL) and alveolar bone, and consequently to gingival recession, deep periodontal pockets, tooth migration, and tooth loss [2].

The ideal therapeutic outcome for PD involves the formation of regularly arranged PDL fibers into regenerated cementum and alveolar bone, ultimately achieving periodontal tissue reconstruction. Although conventional treatments such as guided tissue regeneration can reduce periodontal pocket depth, increase periodontal attachment, and suppress gingival recession, thereby partially improving periodontal conditions, they lack long-term antibacterial capacity and are devoid of stem cells, hence exhibiting limited therapeutic effects, and presenting unpredictable tissue regeneration outcomes in cases of shallow recession defects or thin gingival thickness [3,4]. Consequently, there is an urgent need to develop novel therapeutic strategies to achieve clinically predictable periodontal regeneration.

In recent years, the field of tissue engineering has witnessed continuous advancements, with related technologies holding promise for future clinical applications as innovative strategies for treating periodontal diseases. Scaffolds represent one of the three essential elements in tissue engineering, functioning by regulating a series of key events during periodontal tissue formation to regenerate functional tissue [5,6]. These three-dimensional matrices provide surfaces for cell attachment and proliferation, guide cell differentiation into specific lineages required for periodontal regeneration, and offer mechanical support to maintain structural integrity during tissue repair. Furthermore, some scaffolds can also promote vascular regeneration, regulate the host’s immune response, and exert anti-inflammatory activities, while some scaffolds, such as chitosan scaffolds, also have natural antibacterial properties, which enable them to inhibit pathogenic bacteria and further reduce inflammation, thereby providing a suitable microenvironment for periodontal tissue regeneration [7].

The forms of scaffolds that can promote periodontal regeneration are diverse, such as gels, barrier membranes, fibers, and cell sheets (Figure 1). Barrier membranes have superior space-maintaining ability, and they can prevent the ingrowth of epithelial cells and provide space for the regeneration of alveolar bone and the periodontal ligament. In contrast, fibers can closely mimic the nanofiber network of the natural extracellular matrix. Some researchers have prepared gelatin-hydroxyapatite nanofiber scaffolds via electrospinning technology, and in vitro experiments have shown that these scaffolds have no cytotoxicity to human dental pulp-derived mesenchymal stem cells and can significantly promote their proliferation, thus showing great potential for application in periodontal regeneration. Researchers in another study prepared metronidazole-loaded nanofibers via electrospinning technology as a local drug delivery system for treating PD; the results showed that mechanical treatment combined with metronidazole-loaded nanofibers more effectively inhibited periodontal inflammation than mechanical treatment alone. Moreover, cell sheets can immobilize cells in their self-secreted matrix, preventing the cells from diffusing from the tissue defect area. Researchers in one study prepared autologous PDL cell sheets combined with β-tricalcium phosphate to repair bone defects; the results showed that this combination method can effectively promote the regeneration of severe periodontal defects. However, non-absorbable barrier membranes require a second surgery for removal, while absorbable barrier membranes exhibit unsatisfactory mechanical strength; in contrast, fibers often can only guide cell proliferation and growth on the scaffold surface, with limited guiding ability in three-dimensional space; cell sheets usually can only regenerate simple single-layer tissues; and for periodontal tissue (which contains various complex soft and hard structures), relying solely on cell sheets is not enough [8,9,10,11].

## 2. Hydrogels

Hydrogels are polymeric network systems that can absorb and retain large amounts of water, formed through cross-linking reactions. They are distinguished from other biomedical scaffold materials by their unique structural characteristics and functional properties. The polymeric network of hydrogels can absorb and retain water molecules, and this hydrated state mimics the natural extracellular microenvironment, thereby contributing to their favorable biocompatibility [12]. Hydrogels further exhibit a three-dimensional network structure and viscoelasticity, both of which are analogous to the extracellular matrix (ECM). This enables them to mimic the three-dimensional microenvironment for cellular survival, to provide structural support for cell attachment, and to induce cellular proliferation and lineage-specific differentiation [13]. Moreover, hydrogels possess favorable biodegradability, excellent water/drug absorption capacity, and tunable mechanical properties, making them highly promising for periodontal tissue regeneration [14].

Current drug treatment methods for PD, whether via systemic administration or local use of mouthwashes, struggle to maintain effective therapeutic drug concentrations at the lesion site. Systemic antibiotic administration is also likely to cause dysbiosis and hepatorenal toxicity. Hydrogels enable the direct delivery of antibacterial agents to the periodontal pocket, thereby achieving targeted drug delivery. As hydrophilic three-dimensional network structures, they possess the unique advantages of protecting the activity of drugs and improving encapsulation efficiency, and can locally achieve the slow and controlled release of drugs, prolonging the drug retention time within the periodontal pocket. Consequently, optimal therapeutic drug concentrations can be maintained at the infection site, while this approach reduces systemic side effects [15,16]. Therefore, hydrogels are now regarded as a new therapeutic strategy for treating PD. GSK2606414, a newly identified potent protein kinase RNA-like endoplasmic reticulum kinase (PERK) inhibitor, exhibits therapeutic potential against PD by suppressing endoplasmic reticulum stress. Researchers synthesized GSK2606414-loaded nanoparticles and incorporated them into a hydrogel-based delivery system. The results demonstrated that this drug-loaded hydrogel possessed favorable enzyme-responsive release properties, effectively inhibiting PD progression while promoting alveolar bone regeneration [17].

Hydrogels are broadly classified into natural and synthetic categories. Synthetic hydrogels, such as polyvinyl alcohol (PVA) and polyethylene glycol (PEG), although they can be manufactured through rigorous synthesis processes to produce hydrogels with clear chemical structures, molecular weights, and excellent mechanical strength, do not possess good biodegradability and biocompatibility. Natural hydrogels, including chitosan (CS), alginate (AL), gelatin and hyaluronic acid (HA), exhibit superior biocompatibility and excellent biodegradability, though their mechanical properties are suboptimal [18,19]. Their preparation methods and characterization are shown in Table 1. The limitations of synthetic hydrogels in terms of biocompatibility and biodegradability can be addressed via chemical modification of their functional groups or blending them with natural polymers [20]. In contrast, natural hydrogels such as CS require composite modification with nanoparticles, polymers, and ceramics to enhance their mechanical properties [21]. For instance, Tan et al. [22] prepared a β-tricalcium phosphate (β-TCP)-incorporated hydrogel and found that the addition of this inorganic ceramic enhanced the mechanical properties of the hydrogel. It also simultaneously endowed the hydrogel with excellent chemical, physical, and biological properties while promoting the adhesion, proliferation, and differentiation of osteoblasts. Hu et al. also added hydroxyapatite to hydrogels, and this addition similarly significantly enhanced the hydrogels’ mechanical properties [23]. Therefore, by optimizing chemical structures to improve the respective limitations of synthetic hydrogels and natural hydrogels, developing hydrogels that also possess both excellent biocompatibility and mechanical properties is an important approach to promote the development and clinical application of hydrogels in the future.

Hydrogel development has evolved from traditional to smart systems. Traditional hydrogels primarily form permanent structures through chemical crosslinking networks, but they exhibit significant limitations in practical applications and cannot intelligently respond to external environmental stimuli. Therefore, smart hydrogels—capable of sensing various stimuli (e.g., temperature, pH, pressure, light, and ionic strength)—have emerged [26]. Among smart hydrogels, thermosensitive and photosensitive hydrogels are the most common. Thermosensitive hydrogels can respond to temperature variations and typically possess a lower critical solution temperature, above which they reversibly undergo a sol–gel transition [27]. Arpornmaeklong et al. [28] incorporated β-glycerophosphate into CS/collagen hydrogels to prepare thermosensitive hydrogels. Then, through minimal surgical intervention, they filled irregular tissue defects with the resulting hydrogel in its liquid state. At physiological temperature, this hydrogel underwent a sol–gel transition in response to the thermal stimulation. The mechanism of light-sensitive hydrogels is similar to that of thermosensitive hydrogels, in which the light-sensitive hydrogels respond to light stimulation, This response triggers rapid polymerization of their networks and the formation of multiple tissue interactions, thereby enhancing adhesive strength [29]. Hsia et al. [30] incorporated methacryloyl groups into HA to prepare photo-crosslinkable HA methacryloyl (HAMA) hydrogels. Under visible light irradiation at a wavelength of 405 nm, HAMA underwent crosslinking to form a hydrogel. Meanwhile, by adjusting the solid-to-liquid ratio of the precursor solution, the gelation rate, degradation rate, and physical properties of this hydrogel can be regulated to meet the requirements of tissue engineering.

The responsiveness of smart hydrogels to external physical and chemical factors enables controlled drug release, thereby improving drug bioavailability [31]. Liu et al. [32] developed an intelligent gingipain-responsive hydrogel that responds to gingipain stimuli in the environment to achieve on-demand release of antimicrobial peptides (AMPs). This system significantly inhibited the activity of *Porphyromonas gingivalis* (*P. gingivalis*), effectively controlled PD progression, and promoted the proliferation, migration, and osteogenic differentiation of periodontal ligament stem cells (PDLSCs).

In addition to the above advantages, smart hydrogels can also be tailored into injectable hydrogels for application in periodontal tissue regeneration, that is, filling the tissue defect with the hydrogel precursor solution, which tightly integrates with the surrounding tissue through a sol–gel transition. Injectable hydrogels possess the characteristics of high water content, good controllability, and strong capability for minimally invasive drug delivery [33]. Moreover, their superior flowability enables complete filling of irregular defects in periodontal tissues [34,35].

## 3. Various Active Substances Carried by Hydrogels

Hydrogels can be loaded with substances such as extracellular vesicles (EVs), drugs, seeding cells, and growth factors (Figure 2). Through their unique sustained-release properties, hydrogels can inhibit the growth of periodontal pathogens and modulate host immune responses, thereby treating PD and facilitating periodontal tissue regeneration.

Studies have shown that mesenchymal stem cells (MSCs) can secrete exosomes and other extracellular vesicles. These vesicles can mediate intercellular communication and transport paracrine factors, thereby promoting angiogenesis, immunomodulation, and tissue regeneration [36]. Experimental evidence has confirmed that after MSC-derived EVs were cocultured with chondrocytes, CD73-mediated adenosine in EVs activates Protein Kinase B and extracellular signal-regulated protein kinase (ERK) signaling pathways—both of which promote cell proliferation, migration, and matrix synthesis during cartilage repair [37]. Furthermore, there is evidence suggesting that macrophages can effectively take in the contents of exosomes and alter their own functions, such as triggering macrophage polarization [38]. Macrophages are divided into two major subsets: classically activated (M1) macrophages and alternatively activated (M2) macrophages [39]. M1 macrophages promote T helper 1 (Th1) cell responses, activate osteoclasts and secrete pro-inflammatory cytokines that exacerbate PD and accelerate bone resorption; M2 macrophages, however, secrete anti-inflammatory mediators, regulate the functions of T helper 17 (Th17) cells and regulatory T cells (Tregs), and promote inflammation resolution and tissue regeneration [40]. Dysregulated M1/M2 macrophage polarization is a critical factor in PD-induced periodontal tissue destruction. Su et al. [41] experimentally demonstrated that the secretome of *Porphyromonas gingivalis* (*P. gingivalis*) lipopolysaccharide-pretreated PDLSCs (including EVs) inhibits M1 macrophage polarization, promotes M1-to-M2 macrophage conversion, and induces macrophage migration to inflammatory sites. This finding suggests that stem cell-derived secretions, such as EVs, hold potential for modulating macrophage immune functions and enhancing tissue regeneration.

The combination of antibiotics and mechanical debridement can reduce the number of subgingival periodontal pathogens and may serve as an adjunctive therapy to scaling and root planning. However, the long-term effects of antibiotic use remain unclear, and the issues of antibiotic resistance and the impacts on oral and gut microbiota cannot be overlooked [42]. Antiseptics have a broad antimicrobial spectrum, and they are capable of inhibiting and killing various microorganisms. With diverse and non-specific targets, they are less likely to induce resistance compared to antibiotics [43]. Researchers have developed a CS hydrogel—loaded with triclosan nanoparticles and flurbiprofen—that possesses dual anti-inflammatory and antibacterial effects and demonstrates excellent efficacy in an experimental PD rat model [44]. Moreover, studies have reported loading the antiseptic octenidine into thermosensitive hydrogels and preparing polyhexanide into nanocomposites for the treatment of apical PD and dental caries, respectively, and these two formulations both exhibit excellent biofilm eradication capabilities [45,46]. However, research on the therapeutic effects of the aforementioned two formulations on PD has not been reported, and further in-depth research on this topic is warranted in the future.

To date, researchers have isolated and identified multiple types of MSCs derived from dental tissues, including gingival mesenchymal stem cells (GMSCs), PDLSCs, dental pulp stem cells (DPSCs), stem cells from human exfoliated deciduous teeth (SHED), dental follicle progenitor cells, alveolar bone-derived mesenchymal stem cells, stem cells from apical papilla, and tooth germ progenitor cells. Among these, PDLSCs and GMSCs have been most frequently utilized as seed cells in tissue engineering due to their easy accessibility, self-renewal capacity, multilineage differentiation potential, and immunomodulatory properties. Studies have demonstrated that PDLSCs exhibited higher differentiation potential than GMSCs under osteogenic, adipogenic, and chondrogenic inductive microenvironments, whereas GMSCs showed stronger proliferation capacity compared to PDLSCs, indicating distinct advantages of each cell type [47]. However, when patients have reached advanced stages of periodontal diseases such as aggressive PD, the quality and quantity of autologous periodontal stem cells are often compromised. Bone marrow-derived mesenchymal stem cells (BMSCs) have been shown to secrete cytokines including bone morphogenetic protein-2 (BMP-2), transforming growth factor-beta (TGF-β), fibroblast growth factor-2 (FGF-2), and insulin-like growth factor-1 (IGF-1), all of which can induce cell homing, promote the proliferation and differentiation of local stem cells, and accelerate periodontal tissue regeneration. Additionally, BMSCs have been found to possess intrinsic capabilities to modulate immune responses and suppress inflammation. Consequently, BMSCs have emerged as alternative seed cell sources for periodontal tissue regeneration and have been translated into clinical trials [48]. Pan et al. [49] established a rat alveolar bone defect model to compare new bone formation between GelMA hydrogels encapsulated with human hPDLSCs and the plain GelMA hydrogel controls. Their findings revealed that GelMA hydrogel provided a suitable microenvironment for hPDLSCs adhesion, survival, and osteogenic differentiation. The GelMA + hPDLSCs group demonstrated significantly greater new bone volume, thereby proving the superior bone regeneration efficacy of scaffold-loaded seed cells compared to hydrogel scaffold alone.

Numerous growth factors, including FGF-2, growth differentiation factor-5 (GDF-5), BMPs, and platelet-derived growth factor (PDGF), have been shown to induce cell homing, promote periodontal tissue matrix synthesis and cell differentiation, and play pivotal roles in periodontal tissue regeneration [4,50,51]. Furthermore, studies have shown that TGF-β and interleukin-4 (IL-4) synergize to induce the differentiation of naïve T cells into Tregs, promoting a shift in the immune microenvironment towards an anti-inflammatory state and alleviating periodontal inflammation [52]. In contrast to the EVs described previously, TGF-β primarily modulates the immune microenvironment by regulating T cell immunity, but both can inhibit the progression of PD through immunomodulation. Vascularization and mineralization in bone defect sites remain critical obstacles to overcome in bone regeneration. Divband et al. developed an injectable hydrogel loaded with BMP-2 and vascular endothelial growth factor (VEGF), and they demonstrated that this hydrogel significantly enhanced DPSC proliferation, induced alkaline phosphatase (ALP) gene expression, and promoted vascularization and mineralization in the regenerating bone regions [53].

In addition to commonly used growth factors, researchers have also identified some substances that are equally capable of promoting periodontal tissue regeneration. Integrins are cell surface receptors that mediate mechanical interactions between cells and the ECM and play a critical role in intracellular signaling [54]. Fraser et al. [55] incorporated the integrin-binding peptides Arginine-Glycine-Aspartic acid (RGD) and Gly-Phe-Hyp-Gly-Glu-Arg (GFOGER) into PEG hydrogels, and they demonstrated that RGD and GFOGER directed integrin interactions. Higher concentrations of these peptides enhanced ALP activity, promoted periodontal tissue matrix mineralization, and increased chondrogenic and osteogenic gene expression. Cysteine is frequently present in regulatory, catalytic, and binding sites of various proteins; therefore, the free thiol groups of cysteine incorporated into hydrogels can bind to cell surface proteins and modulate cellular signaling pathways. Hegedűs et al. investigated the effects of thiol group density in hydrogels on cell viability, morphology, and differentiation capacity, and they found that thiol groups improved cell survival, facilitated cell adhesion and proliferation, and simultaneously promoted osteogenic differentiation [56]. Zong et al. [57] pretreated human hPDLSCs with gold nanocomposites (AuNCs) and encapsulated these cells in hydrogels. They found that AuNCs exhibited limited cytotoxicity, were readily internalized by primary hPDLSCs, and enhanced the osteogenic differentiation of hPDLSCs in vitro. Akbarian et al. [58] developed a novel hydrogel incorporating non-immunogenic and non-toxic human serum albumin (HSA) fibrils and found that the presence of these fibrils promoted the attachment of cells including hPDLSCs and directed their migration, proliferation, and metabolism to support tissue regeneration.

Hydrogels loaded with immunomodulators have been regarded as a new method for tumor treatment [59]. The occurrence and development of PD are also inseparable from immune responses. Therefore, hydrogels loaded with different immunomodulators can be used to verify whether they can improve inflammation and promote tissue regeneration by modulating immunity, and to explore which immunomodulator has the best effect. Gene therapy has been used to address skin disease problems such as chronic wounds [60]. Using hydrogels as local gene delivery tools to achieve gene-level repair and regeneration is a highly potential strategy in the field of periodontal regeneration. However, current research on the effects of immunomodulators and gene therapy in promoting periodontal tissue regeneration is relatively scarce, and future studies can focus on these two directions for more in-depth investigation.

Thus, hydrogels can deliver various active substances to treat PD and promote periodontal tissue regeneration. However, different active substances exhibit distinct focuses and effects. EVs primarily act on cells, and they also possess both tissue regenerative and immunomodulatory effects. On the other hand, bactericidal agents and other drugs primarily focus on eliminating periodontal pathogens, thereby reducing inflammation. Seed cells, meanwhile, can directly proliferate, differentiate, and undergo osteogenesis within the three-dimensional microenvironment provided by hydrogels that mimic the natural cellular living environment, thereby directly forming and repairing defective periodontal tissues. Furthermore, growth factors can promote cell proliferation, differentiation, and immunomodulation, and can synergize with seed cells, which plays a crucial role in vascularization and mineralization in bone regeneration areas. Therefore, based on the respective characteristics of the active substances, researchers can select appropriate ones according to actual clinical needs. Additionally, the development of smart hydrogels capable of responding to external stimuli such as temperature, light, pH, and pressure—which enable the autonomous release of loaded substances based on the periodontal microenvironment—can enhance the utilization and efficacy of active substances, thereby better treating PD and promoting periodontal tissue regeneration.

## 4. Recent Advances in the Application of Common Natural Polymer Hydrogels for Periodontal Tissue Regeneration

Currently, CS, AL, gelatin and HA hydrogels have been extensively studied in the field of periodontal tissue regeneration. These natural polymers possess excellent bioactivity and biocompatibility, which contribute to improving scaffold-tissue interactions and enhancing cell adhesion and proliferation [61].

### 4.1. CS Hydrogel

CS, a natural polymer formed by the deacetylation of chitin, is the only cationic polysaccharide in nature. Biomaterials fabricated from CS exhibit excellent biocompatibility, biodegradability, adhesion, and antibacterial properties, and have been extensively utilized in periodontal tissue engineering [62]. CS hydrogels can promote osteogenic differentiation of BMSCs by activating the Wnt3a and ERK1/2 signaling pathways, as well as facilitate chondrogenic differentiation through the Smad2/3 signaling pathway [63]. Studies indicated that CS combined with other polymeric biomaterials and bioceramics (e.g., bioactive glass, calcium phosphate, β-tricalcium phosphate) exhibited greater potential for periodontal regeneration [64]. Suo et al. [65] developed carbon nanotube/CS/AL ternary composite hydrogels with varying concentrations. Test results revealed that these hydrogels possessed favorable mechanical properties and biocompatibility (with the 0.5 wt% carbon nanotube/CS/AL hydrogel scaffolds showing optimal biocompatibility). Additionally, these hydrogel scaffolds promoted PDLSC proliferation and demonstrated antibacterial effects against *P. gingivalis*, potentially attributed to the ability of carbon nanotubes to disrupt bacterial cell membrane integrity.

One study cross-linked CS with inherent antibacterial properties and antimicrobial peptide (AMP) via PEG modification, forming a dual-antibacterial hydrogel CS-PEG-AMP (CS-PA), and subsequently loaded with curcumin-containing biodegradable nanoparticles (CNP). Test results showed that this hydrogel possesses antibacterial and anti-inflammatory effects, and has excellent therapeutic efficacy against PD [66]. In recent years, metal–organic frameworks (MOFs) have received widespread attention in drug delivery and disease treatment. Experiments have demonstrated that magnesium ions can directly promote bone regeneration to some extent, while gallic acid possesses antioxidant and immunomodulatory properties. Therefore, Luo et al [67]. designed an intelligent responsive hydrogel system containing carboxymethyl CS (CMCS), dextran (DEX), and 4-formylphenylboronic acid (4-FPBA), combined with MOFs (including magnesium and gallic acid), and enabled it to release on-demand in the PD environment. Results showed that this system can inhibit inflammation and promote bone regeneration.

Liposomes (Lips), closed phospholipid bilayer vesicles, can encapsulate both hydrophilic and lipophilic drugs [68]. Due to their negative surface charge, Lips can be incorporated into positively charged CS matrices to achieve uniform dispersion throughout the hydrogel system [69]. Atila et al. encapsulated curcumin, a compound with antimicrobial properties, in Lips and subsequently embedded these lips in hydrogels, demonstrating that the composite exhibited remarkable regenerative, antioxidant, and antibacterial properties [70].

Exosomes secreted by stem cells possess anti-inflammatory and immunomodulatory capabilities, enabling the treatment of various diseases. Moreover, since exosomes lack major histocompatibility complex class I or II molecules, they exhibit low immunogenicity [71]. CS has been demonstrated to be an ideal carrier for exosomes. Shen et al. [72] administered CS hydrogels encapsulating DPSC-derived exosomes (DPSC-Exo) to PD-afflicted mice, revealing that DPSC-Exo could be sustainably released from the CS matrix. This process promoted macrophage polarization from pro-inflammatory to anti-inflammatory phenotypes within the periodontal tissue, thereby inhibiting PD progression and enhancing the regeneration of alveolar bone and periodontal epithelium in mice.

Since periodontal tissue defects are often caused by PD, and the occurrence and development of PD are closely related to periodontal pathogens, the inherent antibacterial properties of CS make it particularly suitable for the treatment of PD and periodontal tissue regeneration.

### 4.2. AL Hydrogel

AL-based hydrogels are highly hydrophilic and exhibit excellent biocompatibility. Chenicheri et al. [73] developed an AL/PVA hydrogel for delivering licorice extract, which possesses antimicrobial, antiviral, and immunomodulatory properties. The results showed that this hydrogel could effectively inhibit the growth and survival of major periodontal pathogens. Since melatonin reduces osteoclast activity and inhibits bone resorption by decreasing the expression of Receptor Activator of Nuclear Factor-κB (RANK) mRNA on osteoclast membranes [74], Abdelrasoul et al. [75] fabricated a melatonin-loaded AL/CS/tricalcium phosphate (TCP) composite hydrogel. The findings indicated that this composite hydrogel scaffold accelerated new bone formation, and the generated bone exhibited quality comparable to that of normal bone and favorable periodontal regeneration. In a novel study, researchers utilized sodium AL hydrogels to encapsulate Ag-based MOFs, which exert antibacterial effects by disrupting bacterial intracellular metabolism, generating reactive oxygen species (ROS), compromising cell membrane integrity, and inhibiting biofilm formation. Meanwhile, this hydrogel significantly enhanced endothelial cell invasion and migration capacities, promoted angiogenesis, and downregulated the levels of inflammatory factor levels in periodontal tissues, thus providing a new therapeutic approach for PD [76]. Additionally, given that calcium peroxide (CaO_2_) improves periodontal pocket microenvironments through oxygen release via its reaction with water, while calcium ions (released from CaO_2_) are critical for bone regeneration and ascorbic acid protects cells, injectable CaO_2_ nanoparticle- and ascorbic acid-loaded sodium AL hydrogels suppress anaerobic bacterial growth, reduce inflammation severity, and promote alveolar bone regeneration [77].

Currently, research on AL hydrogels in the field of periodontal tissue regeneration is relatively scarce, and future studies can focus more on this for exploration.

### 4.3. Gelatin Hydrogel

Gelatin, as a natural polymer, has been widely used in hydrogel preparation due to its easy availability and straightforward processability. In addition, its composition is nearly identical to collagen (the main component of the natural ECM) and it contains RGD-like sequences that promote cell adhesion, migration, and wound healing [78]. Studies have demonstrated that transglutaminase (TG) catalyzed the formation of isopeptide bonds between γ-carboxyl groups of glutamine residues and ε-amino groups of lysine residues in gelatin, enabling covalent conjugation of gelatin with tissue amino acids to form stable bonds, thereby enhancing the integration between gelatin-based hydrogel scaffolds and surrounding tissues. Since TG concentration affects hydrogel strength and stability, developing a TG/gelatin composite hydrogel with an optimal concentration of TG may provide a novel approach to address challenges in periodontal tissue regeneration [79].

GelMA is the most prevalent form in gelatin-based hydrogels and it has been extensively applied in biomedical fields. Liu et al. injected GelMA hydrogels loaded with progranulin (PGRN) into experimental dogs with induced PD. The results showed that PGRN suppressed M1 macrophage polarization and Th17 cell differentiation while promoting M2 macrophage polarization and Treg cell differentiation, thereby inhibiting inflammation and modulating the immune response toward an anti-inflammatory state. Furthermore, PGRN enhanced alveolar bone and cementum formation, thus facilitating periodontal tissue regeneration [80].

Building on traditional GelMA hydrogels, researchers have recently developed a series of novel GelMA-based hydrogels. GelMA hydrogels are typically photo-crosslinked via ultraviolet (UV) irradiation in the presence of photoinitiators [81]. However, UV exposure may induce DNA damage in organisms, accelerate tissue aging, and potentially trigger carcinogenesis, posing multiple biological safety concerns [82]. To address this, Goto et al. [83] employed riboflavin and Irgacure 2959 as photoinitiators to fabricate GelMA hydrogels under visible light and UV irradiation, respectively. The results demonstrated that the visible light-polymerized gels exhibited significantly higher viability of encapsulated cells than those polymerized under UV irradiation, while they maintaining suitable properties for osteoblast differentiation. Additionally, researchers have implemented functional group modifications to endow GelMA with intrinsic antibacterial capabilities, thereby enhancing GelMA’s biological performance. Vargas-Alfredo et al. [84] developed a quaternary ammonium-modified GelMA hydrogel possessing anti-inflammatory, antibacterial, and tissue regenerative abilities. This modified hydrogel not only inhibited *P. gingivalis* activity but also promoted human periodontal fibroblast growth. Furthermore, Roldan et al. [85] incorporated biocompatible piezoelectric barium titanate nanoparticle fillers into GelMA prepolymer solutions to create an injectable piezoelectric hydrogel (PiezoGEL). Compared to conventional GelMA hydrogels, PiezoGEL upregulated the expression of osteogenesis-related genes including Runt-related transcription factor 2 (RUNX2), collagen type I alpha 1 chain (COL1A1), and ALP; it also enhanced early ECM mineralization and successfully promoted alveolar bone regeneration while reducing periodontal pocket depth. Simultaneously, PiezoGEL demonstrated anti-inflammatory efficacy by suppressing *P. gingivalis* biofilm formation without pharmaceutical intervention.

GelMA hydrogel has become the most studied type among gelatin hydrogels by scholars currently, and has achieved encouraging results. If it is possible to further study and utilize its photosensitive properties, it will significantly promote the development of smart hydrogels.

### 4.4. HA Hydrogel

HA is a biopolysaccharide and a crucial component of the ECM in connective tissues and the PDL matrix, exhibiting potential for promoting periodontal tissue regeneration. HA regulates cell adhesion, migration, and differentiation by binding to cell membrane proteins such as CD44 [86]. HA hydrogels exist in two distinct forms: hydrogels formed by HA itself and those formed by its salts (e.g., sodium hyaluronate). Data indicated that topical application of HA in nonsurgical periodontal therapy may confer additional clinical benefits [87]. Munar-Bestard et al. [88] incorporated mangostin (MGTN) into HA hydrogels and evaluated their performance. The results demonstrated that MGTN-loaded HA hydrogels significantly inhibited *P. gingivalis* while they exhibited excellent biocompatibility, overcoming the limitations of traditional antiseptics like chlorhexidine, which exhibit high cytotoxicity leading to cell death and tissue damage.

Researchers have improved the properties of HA hydrogels and developed a series of novel HA-based hydrogels. As a natural biomacromolecule, HA is susceptible to degradation by hyaluronidase, which hinders its biomedical applications. Therefore, researchers have utilized 2-chloro-1-methylpyridinium iodide (CMPI) to react with HA, producing an HA auto-crosslinked hydrogel polymer with prolonged in vivo retention and enhanced viscoelasticity [89]. HAMA, formed by the esterification of HA with methacrylic anhydride, is a biocompatible material that enables the photocrosslinking of hydrogels in the presence of Irgacure 2959 [90]. Liu et al. [91] developed a dual-network hydrogel composed of Pluronic F127 (a triblock copolymer) and HAMA, which was loaded with spermidine-modified mesoporous polydopamine nanoparticles (M@S NPs). Experimental results demonstrated that this composite hydrogel combined injectability, thermosensitivity, and photocrosslinking properties; it also alleviated periodontal inflammation by inhibiting the ERK1/2 and NF-κB signaling pathways and exhibited photothermal antibacterial effects, ROS scavenging ability, and anti-inflammatory functions.

HA, as a natural polyelectrolyte, also serves as an electroresponsive drug carrier. Hydrogels fabricated from composites of HAMA, graphene oxide, and polyaniline not only enable electrically triggered drug release but also exhibit excellent mechanical properties, with cumulative drug release enhanced by increased voltage [92]. Additionally, incorporation of copper ions (Cu^2+^) and SHED-derived exosomes (SHED-Exo) into HA hydrogels improves flowability, mechanical strength, and enzymatic resistance. The combined action of Cu^2+^ and HA stimulates macrophages to eliminate periodontal pathogens, while the synergistic effect of SHED-Exo and Cu^2+^ endows hPDLSCs with significant osteogenic activity [18].

As an important component of the ECM, HA has significant advantages in tissue engineering. Solving the problem of its excessively fast degradation rate and further developing HAMA which also has photosensitive properties will be the key development direction for this type of hydrogel next.

The four natural polymer hydrogels exhibit distinct characteristics and unique advantages (Table 2). CS itself exhibits antibacterial properties, but its mechanical properties are somewhat inadequate. Therefore, during preparation, combining CS with synthetic materials such as PEG can be considered to improve its mechanical strength. AL is non-antigenic, but high-molecular-weight AL exhibits poor degradability, and improvements to address this issue are required. Gelatin’s inherent low melting point facilitates the preparation of temperature-sensitive hydrogels, but it also limits its application under physiological conditions. As a major component of the ECM, HA can regulate various intercellular communications and behaviors, has demonstrated considerable potential in in vivo applications, and has gained significant favor among researchers. If the issue of its susceptibility to enzymatic degradation could be resolved while preserving its original properties, this would represent a major breakthrough in the field of periodontal regeneration [24].

Although the experimental results of various hydrogels are encouraging, their clinical application has not yet been successful (Table 3). Due to the complex regeneration process of periodontal tissues, which involves multiple types of tissue cells, it is difficult for a single type of hydrogel to effectively mimic all the characteristics of these tissues. Furthermore, insufficient mechanical strength and insufficient precision of hydrogels in regulating periodontal tissue regeneration are also factors hindering hydrogels’ clinical translation [14]. A clinical study evaluated the effect of HA gel as an adjunctive therapy in periodontal flap surgery for stage III PD, and the results showed that local application of HA gel reduced the content of pro-inflammatory factors and increased the content of anti-inflammatory factors in the patients’ gingival crevicular fluid [93]. However, only a very small number of HA hydrogels are in the transition stage from the preclinical to the clinical research stage, while the vast majority of hydrogels, such as CS, AL, and gelatin, remain in the preclinical research stage. Therefore, addressing the various problems in the clinical translation of hydrogels is the top priority of future research.

## 5. Challenges and Future Directions of Hydrogel Application in Periodontal Tissue Regeneration

Recent years have witnessed significant achievements in hydrogel-related research, which holds promise for addressing challenges in periodontal tissue regeneration. Given their inherent properties, hydrogels possess multiple advantages, including excellent biocompatibility, the ability to mimic the three-dimensional microenvironment for cell survival, support for cell adhesion, induction of cell proliferation and differentiation, favorable biodegradability, unique absorption capacity for biological fluids, and adjustable mechanical properties. Furthermore, hydrogels can serve as drug delivery systems for treating diseases such as PD. With the advancement of research, smart hydrogels responsive to stimuli such as temperature, pH, pressure, light, and ionic strength have been developed. However, studies and applications in this field are still at a relatively preliminary exploratory stage, at which researchers encapsulate various drugs, cells, and growth factors into hydrogels or fabricate multicomponent hydrogels via material blending. Additionally, in-depth investigations into the exploration and optimization of hydrogel structure-property relationships are still lacking. By reviewing the latest literature, the authors have summarized the challenges and potential future development directions in the application of hydrogels for periodontal tissue regeneration (Figure 3). In the future, potential directions in this field may involve designing hydrogels with uniformly porous structures, multilayered systems, and multiphase systems; modulating hydrogel stiffness, swelling properties, and degradation rates to appropriate levels; developing hydrogels with viscoelastic and self-healing properties; and resolving issues related to the hydration surface layer of hydrogels.

### 5.1. Structure of Hydrogels

A porous structure of hydrogels facilitates cell adhesion and proliferation during tissue growth [94]. However, the pore structures of most hydrogels are heterogeneous and disordered, hindering the transport of nutrients and the exchange of gases required for cell proliferation [95]. Since uniform porous structures exhibit superior performance in promoting cell growth [96], researchers aim to fabricate hydrogels with high porosity and uniformly sized pores. Xu et al. utilized nanoparticles with small size effects and surface effects as crosslinking agents to address the issue of irregular crosslinking in collagen induced by traditional crosslinking agents, thereby fabricating hydrogels with more uniform pore structures [97]. As a foaming agent, magnesium can generate gas during its degradation process to promote pore formation, optimize porosity, and enhance vascularization and bone regeneration [98]. Additionally, microfluidic technology allows for precise control over the hydrogel fabrication process to obtain hydrogels with homogeneous pore architectures [99]. As an emerging biomaterial, microporous annealed particle (MAP) hydrogels exhibit potential for promoting tissue repair and regeneration. By employing bioorthogonal tetrazine click chemistry to anneal PEG hydrogel microparticles into MAP hydrogels in situ, intrinsic microporous structures can be formed at tissue defect sites, improving pore uniformity in hydrogels [100]. Moreover, 3D bioprinting technology utilizes biomaterials or cells as “bioinks” to fabricate precise, biomimetic, and personalized scaffolds, allowing researchers to flexibly design scaffold shape, size, pore structure, and porosity according to specific needs [101]. Scholars conducted in vitro performance evaluations on hydrogels prepared via 3D bioprinting technology, and the results showed that these 3D-printed scaffolds exhibited uniform porous structures and good swelling properties [102]. In recent years, the concept of 4D printing has been further proposed. On the basis of 3D printing, 4D printing introduces the time dimension, through external stimuli such as pressure, air, heat, water acting on smart materials, enabling the materials to possess the ability to change shape over time [103]. This characteristic enables it to make adaptive changes according to the specific conditions of the periodontal microenvironment, being more intelligent and efficient.

Since periodontal tissue is a complex tissue composed of multiple structures, designing multilayered hydrogels with distinct materials, densities, and functions in each layer can effectively mimic the intricate periodontal environment and promote periodontal tissue regeneration. Recently, scaffolds containing both dense and porous layers have demonstrated potential for enhancing periodontal tissue regeneration, in which the dense layer effectively isolates bone defects from surrounding connective tissues, while the porous layer facilitates the regeneration of bone and PDL [104]. Researchers designed a three-layered nanocomposite hydrogel scaffold, in which each layer (composed of different materials and growth factors) was assembled according to the positional sequence of periodontal tissues, achieving simultaneous regeneration of cementum, PDL, and alveolar bone [105]. Furthermore, the multiphasic nature is another structural characteristic desired in hydrogels. This type of scaffold can provide a well-compartmentalized environment, enabling periodontal progenitor cells to undergo guided differentiation into diverse lineages, thereby allowing them to regenerate various periodontal tissues [106].

Taken together, compared to traditional hydrogels, novel hydrogels with uniformly porous structures, multilayered, and multiphasic systems can more effectively promote periodontal tissue regeneration. They are expected to address challenges unresolved by conventional scaffolds and advance to clinical trials; furthermore, bioprinting represents a potential approach to achieve these objectives. However, due to the inherent complexity and variability of hydrogels and cells, significant challenges remain in fabricating scaffolds that precisely meet pre-designed specifications [107]. Researchers have developed a 3D-printed composite hydrogel composed of CS, gelatin, and baicalin extract. They set up three concentration groups, 2% (*w*/*v*) CS, 2.5% (*w*/*v*) CS, and 3% (*w*/*v*) CS, and conducted structural and performance tests on the hydrogels of these groups. The results indicated that higher CS concentrations led to greater viscosity, while the hydrogel with 2.5% (*w*/*v*) CS exhibited moderate ink viscosity, ensuring good printability. When using a G22 nozzle, a printing speed of 25 mm/s, and an approximate printing pressure of 85 kPa, continuous and uniform extrusion was achieved at 25 °C, ultimately resulting in scaffolds with high shape fidelity, excellent swelling properties, and controllable drug release [78]. Thus, ideal printing parameters require a balance between material properties and printing parameters. Therefore, based on the characteristics of the developed material, it is necessary to conduct grouped experiments on factors affecting printing outcomes—such as bioink concentration, nozzle size, printing pressure, printing speed, and printing temperature—so as to ensure moderate ink viscosity and other critical printing conditions, and thereby obtain optimal printing parameters.

Thus, identifying optimal printing parameters through extensive experimental data and digitally constructing hydrogel structures with uniformly porous, multilayered, and multiphasic systems will be a key research focus moving forward.

### 5.2. Performance of Hydrogels

The mechanical properties of hydrogels play a critical role in bone regeneration. High-strength scaffolds can upregulate the expression of the epigenetic regulator ten-eleven translocation methylcytosine dioxygenase 2(TET2), and subsequently suppress E-cadherin transcription, promote sustained activation of the Wnt/β-catenin pathway, and enhance the osteogenic capacity of PDLSCs [108]. In addition, the study found that all matrix metalloproteinase (MMP)-degradable hydrogels increased the ALP activity of PDLC. Among them, softer hydrogels could further enhance the ALP activity of PDLC, but their matrix mineralization level was lower, while matrix mineralization was maximized in stiffer hydrogels. This indicates that increased ALP activity cannot directly lead to matrix mineralization, and the latter is also strictly dependent on the stiffness of the hydrogels. These experimental results reflect the complexity of PDLC’s response to ECM signals and the limitations of current scaffold materials [109]. Research indicated that cells continued to exhibit physiological behaviors influenced by the mechanical properties of their original ECM even when transferred to different ECM environments, a phenomenon termed “mechanical memory”. The conversion of hydrogels’ mechanical properties into biological behaviors is closely associated with intracellular signal transduction [110]. The Yes-associated protein has been identified as a mechanosensor capable of perceiving mechanical stimuli and enhancing the secretion of osteogenic markers such as ALP [111]. This also explains why the stiffness of hydrogels can influence the osteogenic differentiation of stem cells, and this indicates that the proliferation, differentiation, and osteogenesis of stem cells can be modulated by tailoring the stiffness of hydrogels. Talaei et al. [112] employed response surface methodology to screen for optimal compositions of GelMA/HAMA hybrid hydrogel systems, significantly reducing experimental iterations while identifying concentrations that effectively improve mechanical properties. This statistical approach could also be applied to other hydrogels to determine their optimal compositions or crosslinking conditions. It is noteworthy that there is a negative correlation between the previously mentioned porosity and the mechanical strength of hydrogels; as the porosity increases, the mechanical strength of hydrogels tends to decrease [79]. However, simultaneously optimizing cell growth and osteogenic differentiation requires hydrogels to possess both high mechanical strength and high porosity. Based on this research challenge, Evans et al. developed a novel hydrogel system based on dendronized polymers, which can independently regulate the mechanical properties and porosity of the hydrogel, decoupling the interrelationship between structure and properties during its preparation, to better adapt to the needs of tissue regeneration [113].

PDLC alignment can be controlled by modulating the swelling of hydrogels. Additionally, in composite hydrogels with minimal swelling capacity, PDLCs exhibit random alignment near the dentin, whereas hydrogels with higher swelling capacity induced PDLC alignment perpendicular to the dentin surface. Furthermore, the degree of alignment and extension increases with enhanced swelling [114]. Therefore, modulating the swelling capacity of hydrogels facilitates control over PDLC alignment and extension direction during periodontal tissue regeneration.

The degradation rate of hydrogels also influences the efficacy of periodontal tissue regeneration. However, for certain types of hydrogels, the human body lacks enzymes capable of degrading their matrix, which restricts the degradation of hydrogels, is unfavorable for the release of stem cells at the defect site, and urgently requires finding effective strategies to solve the problem of controllable degradation. Recent research demonstrated that hydrogel degradation rates could be modulated through enzymatic catalysis, ester hydrolysis, photolysis, or incorporation of oxidizing agents and fibrinogen [115]. Gohil et al. [116] developed a novel enzyme-sensitive hydrogel by acetylating hydroxypropyl pyrrolidone-grafted CS (HPP-GC) hydrogels. Its degradation rate could be adjusted by adjusting the degree of acetylation, while in vivo degradation rates and release profiles were tunable across a broad range. 19F magnetic resonance imaging (19F-MRI) enables real-time, non-invasive tracking, precise localization, and accurate quantification of in vivo hydrogels degradation rates, thereby facilitating the assessment of degradation rates, the subsequent investigation of influencing factors, and the optimization of hydrogel performance [117].

Studies show that the viscoelasticity of hydrogels can regulate the chondrogenesis of MSCs in a Rho-associated kinase-dependent manner; slower relaxing hydrogels are beneficial for the initiation of chondrogenic differentiation but detrimental to long-term chondrogenesis; faster relaxing hydrogels, although beneficial for long-term chondrogenesis, are detrimental to early chondrogenic differentiation [118]. However, most existing hydrogel scaffolds fail to replicate the native viscoelastic properties of periodontal tissues. Therefore, researchers specifically designed a dual-crosslinked viscoelastic scaffold and established a rat model for experimentation. The results demonstrated that this viscoelastic hydrogel significantly promoted periodontal collagen deposition at the injury site and accelerated periodontal injury repair [119]. Moving forward, how to regulate the viscoelastic properties of various hydrogels should be regarded as the main research focus.

The extrusion pressure of injectable hydrogel delivery devices and the mechanical forces from limb movement often damage hydrogels, leading to rapid degradation of the hydrogel matrix and leakage of loaded therapeutic substances. Hydrogels crosslinked via dynamic covalent crosslinking possess self-healing properties, enabling automatic repair of structural defects and functional recovery, thus addressing the aforementioned issues [120]. Guo et al. designed a dual-dynamic crosslinking network (DDCN) hydrogel by utilizing dynamic Schiff base bonds and dynamic coordination bonds, which exhibits rapid gelation, injectability, and excellent self-healing ability [121].

It is noteworthy that hydrogels share a common drawback: the presence of a hydrated surface layer suppresses the adsorption of ECM and specific proteins, which impairs cellular activity. This undermines the inherent advantages of hydrogels as scaffolds, is unfavorable for cell attachment, proliferation, and differentiation, and affects the tissue regeneration outcome. Studies have shown that altering the nature or density of hydrogel’s charge can regulate their interactions with biomolecules and influence cell attachment. For example, negatively charged hydrogels stimulate the attachment and proliferation of chondrocytes and endothelial cells, whereas positively charged hydrogels promote the attachment of osteoblasts, neuronal cells, and fibroblasts [122]. Moving forward, further research is needed to explore the impact of charge density on hydrogel–biomolecule interactions and to identify alternative strategies addressing the challenge of hydrated surface layers.

Therefore, several areas for further improvement of hydrogels in terms of their structural and performance characteristics include designing hydrogels with uniformly porous structures, multilayered and multiphase systems; modulating hydrogels’ stiffness, swelling properties, degradation rates and viscoelasticity; developing hydrogels with self-healing properties; and addressing challenges such as the hydrated surface layer issue.

## 6. Summary and Prospects

Hydrogels have been widely applied in periodontal tissue regeneration due to their excellent biocompatibility, biodegradability, injectability, ECM-mimicking capability, and sustained drug release properties. Researchers, based on hydrogels such as CS, AL, gelatin, HA, etc., loaded them with different drugs, seed cells, growth factors, and other substances, tested their comprehensive effects on periodontal tissue regeneration, and achieved encouraging results. However, current hydrogels still exhibit numerous limitations. This review comprehensively analyzes the relevant literature to summarize aspects of hydrogels that require urgent improvement, from both structural and performance perspectives.

Since periodontal tissue is a functional complex composed of cementum, PDL, and alveolar bone, it remains to be explored whether multigradient hydrogels mimicking natural periodontal tissue can be fabricated for these distinct structures to achieve efficient periodontal regeneration. The insufficient mechanical properties of hydrogels represent a major barrier to their clinical translation. Therefore, attempts can be made to incorporate new polymers without inducing cytotoxicity to address this issue. Another challenge hindering the clinical translation of hydrogels is their precision in regulating periodontal tissue regeneration. It is possible to utilize 4D printing to pre-set the spatiotemporal changes of hydrogel shape and other properties, enabling it to complete specific tasks at a specific time according to the specific conditions of the periodontal microenvironment, perhaps successfully regenerating the desired periodontal tissue.

Periodontal inflammation is a primary cause of periodontal tissue defects. Before regenerating tissues, controlling inflammation should be prioritized. Therefore, hydrogel-based treatments should be divided into two stages: first, using hydrogels to deliver antibacterial and anti-inflammatory drugs to inhibit the activity of periodontal pathogens and reduce inflammation; and after inflammation stabilizes, the treatment switches to drugs or active substances that promote periodontal tissue regeneration. However, repeated injections of hydrogels may lead to poor patient compliance. Multi-structured composite hydrogels can be designed to carry both types of drugs simultaneously, and through 4D printing or intelligent responsiveness, they can enable the controlled release of different drugs over time and in response to environmental changes.

Currently, researchers have conducted extensive studies on the effects of hydrogels loaded with drugs, stem cells, and other active substances for treating periodontal inflammation and promoting periodontal tissue regeneration, and have achieved promising results. However, we can perhaps try other types of substances, such as the aforementioned immunomodulators, gene delivery, etc.

Furthermore, establishing a systematic evaluation system for the effects of hydrogels on promoting periodontal tissue regeneration, constructing statistical models, collecting extensive experimental data, and integrating these data to establish a quantitative relationship between hydrogel properties and regenerative outcomes can provide guidance for researchers on which components to add during hydrogel preparation and at what concentrations, thereby improving research efficiency. Additionally, the 3D and 4D printing technologies that have developed rapidly in recent years should be effectively utilized to improve the structure and properties of hydrogels, and to develop precise and personalized biomimetic scaffolds tailored to individual patient characteristics.

In the future, effectively addressing the challenges in periodontal regeneration through targeted hydrogel modifications will be the top priority for scientific and clinical research.

## Figures and Tables

**Figure 1 pharmaceutics-17-01382-f001:**
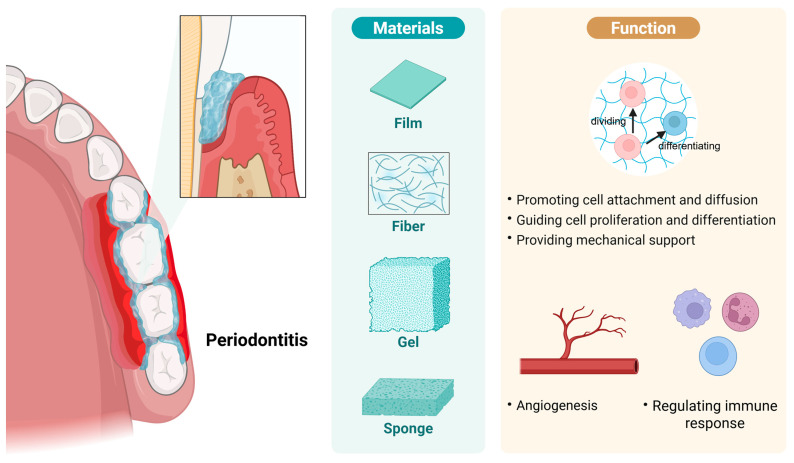
Common scaffold types and their roles in periodontal tissue regeneration. In the process of treating periodontitis and promoting tissue regeneration, various forms of scaffolds are often employed, including barrier membranes, fibers, gels, and cell sheets. Scaffolds can promote cell attachment and spreading, guide cell proliferation and differentiation, and provide mechanical support for regenerating biological structures. Furthermore, certain types of scaffolds can promote vascular regeneration, modulate the host immune response, thereby alleviating inflammation, and inhibit the growth of pathogenic bacteria. Created in https://BioRender.com (accessed on 20 August 2025).

**Figure 2 pharmaceutics-17-01382-f002:**
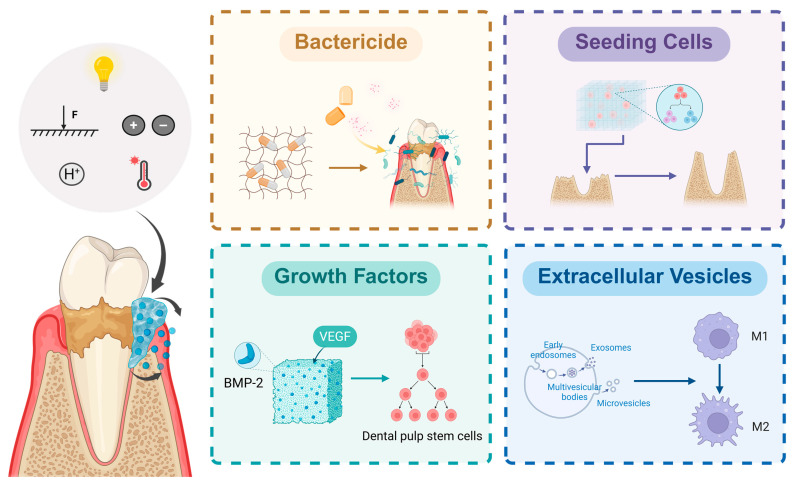
Active substances loaded in hydrogels and their applications. Intelligent hydrogels responsive to external factors such as temperature, light, pH, and pressure can autonomously release loaded active substances according to the periodontal microenvironment. Different active substances exert distinct application effects. Disinfectants can inhibit the growth and survival of periodontal pathogens and alleviate inflammation; seed cells can directly proliferate, differentiate, and undergo osteogenesis to form and repair defective periodontal tissues; growth factors can promote cell proliferation and differentiation; and extracellular vesicles possess immunomodulatory functions. Created in https://BioRender.com (accessed on 20 August 2025). Abbreviations: F: Force.

**Figure 3 pharmaceutics-17-01382-f003:**
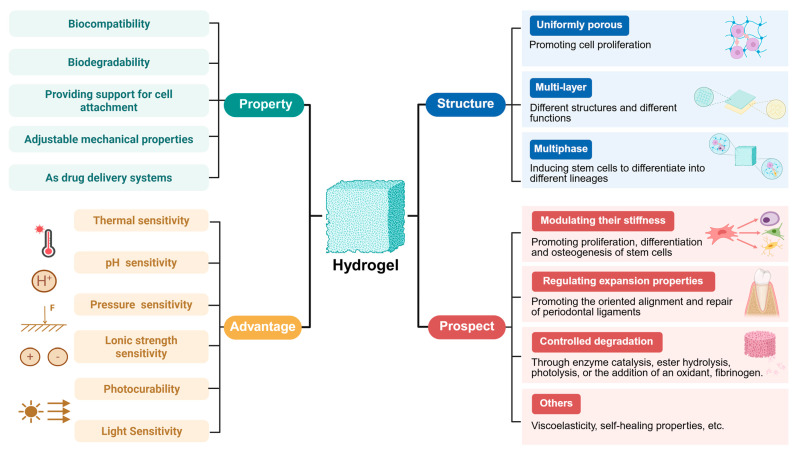
Characteristics, advantages, and future prospects of hydrogels for periodontal tissue regeneration. Created in https://BioRender.com (accessed on 20 August 2025).

**Table 1 pharmaceutics-17-01382-t001:** Basic Information, Preparation Methods, and Characterization of Common Hydrogels.

Type	Basic Information	Preparation Method	Characterization	References
Chitosan Hydrogel	Composed of repeating residues of D-glucosamine and N-acetyl-D-glucosamine	Physical crosslinking: Forms a gel network through non-covalent interactions (electrostatic interactions, etc.); Chemical crosslinking: Crosslinking is performed using bifunctional/multifunctional molecules	High biocompatibility; Insufficient mechanical strength; The degradation rate is not ideal	[24]
Alginate Hydrogel	Composed of repeating residues of α-L-guluronic acid (G) and β-D-mannuronic acid (M)	Ionic crosslinking: Ionic crosslinking is performed with divalent cations (e.g., Ca^2+^, Ba^2+^)	High biocompatibility; The mechanical properties are related to the crosslinking cation concentration, G-block length, etc.; High-molecular-weight alginate has a low degradation rate	[24]
Gelatin Hydrogel	Denatured water-soluble polypeptides obtained from irreversible hydrolysis of collagen	Gelatin aqueous solutions at 0.5–50 wt% are in a sol state above the melting point and form a thermoreversible gel upon cooling	Good biocompatibility and biodegradability	[24]
Hyaluronic Acid Hydrogel	Composed of N-acetylglucosamine and D-glucuronic acid residues	Self-crosslinking is achieved through the formation of intramolecular and intermolecular ester bonds between carboxyl and hydroxyl groups	High biocompatibility; The degradation rate in vivo is extremely fast	[24]
Polyvinyl Alcohol Hydrogel	Synthetic Polymer Hydrogel	Physical crosslinking: Based on non-covalent interactions such as hydrogen bonding; Chemical crosslinking	Good mechanical properties; Degradability: Presents low biodegradability issues	[25]
Polyethylene Glycol Hydrogel	Synthetic Polymer Hydrogel	Primarily prepared via chemical crosslinking methods	Good mechanical properties	[25]

**Table 2 pharmaceutics-17-01382-t002:** The advantages and disadvantages of common natural polymer hydrogels.

Type	Advantages	Disadvantages
Chitosan Hydrogel	Possessing intrinsic antibacterial properties	Insufficient mechanical properties; Suboptimal degradation rate; poor solubility
Alginate Hydrogel	Being non-antigenic; Capable of forming gel under mild conditions; Possessing long-term stability	High-molecular-weight alginate is difficult to completely clear in vivo
Gelatin Hydrogel	Capable of forming thermoreversible gel	Low melting point limits its application under physiological conditions.
Hyaluronic Acid Hydrogel	a core component of the extracellular matrix, capable of regulating intercellular communication and behavior	Degrades too rapidly in vivo; pure hyaluronic acid hydrogel exhibits poor stability.

Note: This table is summarized based on the article by Zhao et al. [24].

**Table 3 pharmaceutics-17-01382-t003:** Representative studies and translation level of common natural polymer hydrogels.

Type	Representative Studies			Translation Level	References
	Authors	Method	Result		
Chitosan Hydrogel	Suo et al. (2023) [65]	Prepared carbon nanotube/chitosan/alginate ternary composite hydrogel at a concentration of 0.5%	Promoted PDLSC proliferation and inhibited *Porphyromonas gingivalis* growth	Preclinical research	[14]
	Xu et al. (2023) [66]	Chitosan was crosslinked with antimicrobial peptides via polyethylene glycol modification to form a dual-antibacterial hydrogel, which was then loaded with curcumin-containing nanoparticles	Possessed antibacterial and anti-inflammatory effects, and exhibited excellent efficacy against periodontitis		
	Shen et al. (2020) [72]	Prepared dental pulp stem cell-derived exosome chitosan hydrogel and injected it into periodontitis mice.	Inhibited periodontitis in mice and promoted the healing of alveolar bone and periodontal epithelium in mice		
Alginate Hydrogel	Chenicheri et al. (2022) [73]	Prepared alginate/polyvinyl alcohol hydrogel and loaded with licorice	Inhibited the growth and survival of major periodontal pathogens	Preclinical research	[14]
	Abdelrasoul et al. (2023) [75]	Prepared alginate/polyvinyl alcohol hydrogel and loaded with licorice	Promoted new bone formation, and the new bone quality was similar to normal bone		
Gelatin Hydrogel	Liu et al. (2024) [80]	Gelatin methacryloyl hydrogel loaded with particulate protease precursor was injected into periodontitis dogs	Inhibited inflammation and promoted alveolar bone and cementum formation	Preclinical research	[14]
	Roldan et al. (2023) [85]	Gelatin methacryloyl hydrogel was combined with biocompatible piezoelectric filler barium titanate to develop an injectable piezoelectric hydrogel	Compared with traditional gelatin methacryloyl hydrogel, upregulated osteogenesis-related gene expression and inhibited *Porphyromonas gingivalis* biofilm formation		
Hyaluronic Acid Hydrogel	Munar-Bestard et al. (2024) [88]	Loading mangostin into hyaluronic acid hydrogel	Significantly inhibited *Porphyromonas gingivalis*	Primarily preclinical research; Few clinical studies	[14,93]
	Liu et al. (2024) [91]	Prepared a dual-network hydrogel composed of Pluronic F127 and methacrylated hyaluronic acid, and loaded with spermidine-modified mesoporous polydopamine nanoparticles	Possessed photothermal antibacterial, reactive oxygen species scavenging, and anti-inflammatory effects		

## Data Availability

Data sharing is not applicable to this review as no new data were created or analyzed in this study.

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
