# Peer review of "Hydrogel-Based Therapeutic Strategies for Periodontal Tissue Regeneration: Advances, Challenges, and Future Perspectives"

_pharmaceutics, 2025, doi:10.3390/pharmaceutics17111382_

Round 1
Reviewer 1 Report
Comments and Suggestions for Authors
This review manuscript provides a timely and comprehensive overview of hydrogel-based approaches for periodontal tissue regeneration. It covers key aspects including the characteristics of hydrogels, various bioactive substances they can carry, recent applications of specific hydrogel types (e.g., chitosan, alginate, gelatin, and hyaluronic acid), and future challenges and directions. The topic is highly relevant to the field of pharmaceutics, particularly in drug delivery systems and tissue engineering scaffolds for oral health applications. The authors effectively highlight the transition from traditional to smart hydrogels and emphasize their potential in addressing limitations of conventional periodontitis treatments. The inclusion of figures (e.g., scaffold types, bioactive substances, and future prospects) enhances readability and illustrates concepts well.
The manuscript is well-structured, with logical progression from introduction to challenges. It synthesizes recent literature (up to 2025) and proposes forward-looking solutions, such as uniform porous structures, multilayered designs, and performance optimizations like tunable stiffness and degradation rates. This makes it a valuable resource for researchers in biomaterials and regenerative dentistry.
However, there are areas for improvement, including language polishing for clarity and flow, deeper critical analysis in some sections, and minor factual clarifications. Overall, the manuscript is suitable for publication with minor revisions.
Strengths
1. Comprehensive Coverage: The review adeptly summarizes the role of hydrogels as both drug delivery platforms and tissue engineering scaffolds. Sections on active substances (e.g., EVs, antibiotics, seed cells, growth factors) and specific hydrogel types (CS, AL, gelatin, HA) are thorough, with relevant examples from recent studies (e.g., GSK2606414-loaded hydrogels, DPSC-Exo in CS matrices).
2. Balanced Discussion of Advances and Challenges: The authors critically examine limitations in clinical translation, such as mechanical properties, degradation rates, and hydrated surface layers, while proposing practical solutions (e.g., 3D bioprinting for uniform pores, statistical modeling for optimization). This adds depth and forward-thinking perspective.
3. Up-to-Date References: The bibliography includes recent publications (e.g., 2024-2025), ensuring the review reflects current trends like smart hydrogels and exosome integration.
4. Visual Aids: Figures 1-3 are clear, informative, and created using BioRender, which supports reproducibility. They effectively summarize complex ideas, such as scaffold roles and hydrogel prospects.
5. Relevance to Journal Scope: Aligns well with Pharmaceutics’ focus on drug delivery, biomaterials, and therapeutic strategies, especially for controlled release in oral applications.
Major Concerns
None identified. The content is scientifically sound, with no major factual errors or gaps that undermine the review’s validity. However, the discussion could benefit from more critical synthesis rather than descriptive listing in sections like 4 (applications of hydrogel types), where studies are summarized sequentially without strong comparative analysis (e.g., which hydrogel type shows superior efficacy in vivo?).
Minor Concerns
1. Language and Clarity: The English is generally good but occasionally awkward or repetitive. For example:
• Page 2, line 41: “urgent need to develop novel therapeutic strategies to achieve clinically predictable and sustainable periodontal regeneration.” – Consider rephrasing to avoid redundancy (e.g., “clinically predictable periodontal regeneration”).
• Page 3, line 85: “mild cytotoxicity and antigenicity, which may lead to immune rejection” – Specify if this is for all synthetic hydrogels or certain ones.
• Page 5, line 157: “Studies have demonstrated that EVs, including exosomes secreted by mesenchymal stem cells (MSCs) and dental pulp stem cells (DPSCs), can mediate intercellular communication…” – This sentence is long; break it for better flow.
• Proofread for minor grammatical issues, e.g., inconsistent capitalization in section titles (e.g., “The characteristics of hydrogels” vs. “Various active substances carried by hydrogels”).
2. Depth of Analysis: In section 3 (active substances), add brief comparisons of efficacy (e.g., EVs vs. growth factors in immunomodulation). In section 5 (challenges), expand on how 3D printing parameters could be optimized with specific examples from literature.
3. Figures and Tables: Figures are well-integrated, but consider adding a table summarizing key hydrogel types, their pros/cons, and representative studies for quick reference.
4. References: Ensure all citations are accurate; some 2025 references (e.g., ref. 67, 89) appear forward-dated but are acceptable if preprints or accepted manuscripts. Check for formatting consistency (e.g., journal abbreviations).
5. Author Contributions and Conflicts: The contributions section is clear, but explicitly state if funding influenced the review (though no conflicts are declared).
Reviewer 2 Report
Comments and Suggestions for Authors The authors described four types of hydrogels that could be used in periodontal tissue regeneration. The article is well-prepared, but it has several shortcomings that should be corrected before publication: 1. The authors focused on several functions of materials for use in periodontitis, for example, in Figure 1. However, considering the pathogenesis of periodontal diseases, the most important are the antimicrobial and anti-inflammatory effects. This is missing from both the figure and the text. 2. Chapter 3 shows the use of antibiotics in hydrogels. However, currently, a campaign is underway to protect antibiotics. Therefore, local antibiotic use is not recommended. Antiseptics, such as octenidine and polihexanide, are used instead. Unfortunately, this topic is not described, although it is essential. The authors only mention the cytotoxic chlorhexidine. 3. Figure 3 is small and not very visible.Author Response
Please see the attachment.

Reviewer 3 Report
Comments and Suggestions for Authors
This review addresses an important and timely topic—hydrogel-based strategies for periodontal tissue regeneration. The manuscript is comprehensive, covering natural, synthetic, and smart hydrogels, as well as their applications as scaffolds and drug delivery systems. The topic fits well within the scope of Pharmaceutics and is relevant for both biomaterials researchers and clinicians.
However, the manuscript requires major revision before it can be considered for publication. The main concerns include:
-
Critical depth – The review is largely descriptive, with limited analysis of limitations, conflicting results, and translational challenges. More critical evaluation is necessary.
-
Comparative synthesis – A table or figure comparing hydrogel types (mechanical properties, biodegradability, biocompatibility, translational status) would add significant value.
-
Clinical perspective – The discussion should better differentiate between preclinical studies and clinical evidence, highlighting barriers to translation.
-
Conciseness and clarity – Some sections, especially the Introduction and hydrogel classification, are repetitive and would benefit from streamlining.
-
Figures – Current figures are useful but mainly illustrative; integrative schematics linking hydrogel categories, active agents, and clinical outcomes are recommended.
-
References – The reference list is extensive but contains duplicates, redundant citations, and inconsistent formatting. The authors should consolidate overlapping references, prioritize recent high-impact studies, and ensure compliance with journal style.
Reviewer 4 Report
Comments and Suggestions for Authors
Introduction: The author should add a description of the hydrogel, why hydrogels have been used to manage PD. Moreover, the unique properties and advantages of hydrogels are also included. Further, the hypothesis and flow of the article should be outlined. Additionally, more literature should be included related to other scaffolds used to manage the disease, along with the pros and cons.
Heading 2: It should be hydrogels: In this section, the descriptive details about the hydrogels, their preparation method and their characterisation should be added. It is recommended that these data be added to the table form.
The tile mentions hydrogel-based strategies, but heading 4 presents different polymer-based hydrogels. Therefore, it is suggested that either modify the title or include the various strategies used for the management of PD. Moreover, the literature should be compiled in a table form. Additionally, section 4 should contain some good quality figures presenting the application of hydrogels used to manage the disease.
Comments on the Quality of English LanguageThe manuscript should be thoroughly checked for grammatical errors and syntax errors
Round 2
Reviewer 2 Report
Comments and Suggestions for Authors
The authors significantly corrected the manuscript according to the reviewer's suggestions. Recently, I recommend the article for publication.
Author Response
We sincerely thank you for your positive feedback and valuable comments, which have significantly improved our manuscript. We are pleased that the revisions have addressed all the points you raised.
Reviewer 3 Report
Comments and Suggestions for Authors
-
ine 353–354: “dual-antibacterial hydrogels (CS-PA)” — “PA” is not defined.
-
Line 359: “metal-organic frameworks (containing magnesium and gallic acid)” — magnesium is not typical in MOFs; may need clarification.
-
Line 403–404: “suppress anaerobic bacterial colonization” — “colonization” should be “growth” or “proliferation”.
-
Line 624–628: Confusing sentence: “In softer… hydrogels, ALP activity is higher, yet this does not necessarily promote matrix mineralization…” — unclear whether this is a contradiction or context-dependent.
-
Line 667–669: “viscoelasticity of hydrogels can coordinate mesenchymal stem cell (MSC) chondrogenesis and survival in a Rho-associated kinase-dependent manner” — “coordinate” is vague.
-
Line 117–118: Claims synthetic hydrogels lack biodegradability and biocompatibility, but later (Line 144–146) suggests blending natural and synthetic polymers to overcome these issues — slightly contradictory without clarification.
-
Line 353–357: Describes “CS-PA” but does not define “PA” — likely “antimicrobial peptide” but not specified.
-
Line 624–628: Presents conflicting findings on stiffness and mineralization without resolving the contradiction.
-
Line 684–693: Discusses “hydrated surface layer” as a drawback, but does not connect clearly to prior sections on hydrogel advantages.
-
Sections 4.1–4.4: Each subsection ends abruptly; no transitional sentences to the next.
-
Section 5.1 and 5.2: Discuss structure and performance separately, but some content overlaps (e.g., porosity affects mechanical properties).
-
Conclusion (Section 6): Introduces new ideas (e.g., 4D printing, immunomodulators) that were not discussed in earlier sections.
-
Line 101“echanical” → “mechanical”.
-
“F” in “Antiseptics” diagram is unexplained.
-
Line 625: “MMP-degradable hydrogels” is used without being defined earlier.
Reviewer 4 Report
Comments and Suggestions for Authors
Now, the manuscript is ok. So please accept it.
Author Response

(The authors gave the same response as above.)

Round 3
Reviewer 3 Report
Comments and Suggestions for Authors
Accept in present form